# Antimalarial Mechanisms and Resistance Status of Artemisinin and Its Derivatives

**DOI:** 10.3390/tropicalmed9090223

**Published:** 2024-09-20

**Authors:** Dan Zheng, Tingting Liu, Shasha Yu, Zhilong Liu, Jing Wang, Ying Wang

**Affiliations:** 1Department of Tropical Medicine, College of Military Preventive Medicine, Army Medical University, Chongqing 400038, China; zhengyangd679@163.com (D.Z.); liutingting@tmmu.edu.cn (T.L.); shasha102@126.com (S.Y.); liuzhilong@tmmu.edu.cn (Z.L.); wjtc1024102410@163.com (J.W.); 2School of Public Health, The Key Laboratory of Environmental Pollution Monitoring and Disease Control, Ministry of Education, Guizhou Medical University, Guiyang 561113, China

**Keywords:** artemisinin, malaria, mechanism of action, *Plasmodium*, resistance, kelch13

## Abstract

Artemisinin is an endoperoxide sesquiterpene lactone isolated from *Artemisia annua* and is often used to treat malaria. Artemisinin’s peroxide bridge is the key structure behind its antimalarial action. Scientists have created dihydroartemisinin, artemether, artesunate, and other derivatives preserving artemisinin’s peroxide bridge to increase its clinical utility value. Artemisinin compounds exhibit excellent efficacy, quick action, and minimal toxicity in malaria treatment and have greatly contributed to malaria control. With the wide and unreasonable application of artemisinin-based medicines, malaria parasites have developed artemisinin resistance, making malaria prevention and control increasingly challenging. Artemisinin-resistant *Plasmodium* strains have been found in many countries and regions. The mechanisms of antimalarials and artemisinin resistance are not well understood, making malaria prevention and control a serious challenge. Understanding the antimalarial and resistance mechanisms of artemisinin drugs helps develop novel antimalarials and guides the rational application of antimalarials to avoid the spread of resistance, which is conducive to malaria control and elimination efforts. This review will discuss the antimalarial mechanisms and resistance status of artemisinin and its derivatives, which will provide a reference for avoiding drug resistance and the research and development of new antimalarial drugs.

## 1. Introduction

Malaria is an infectious disease with the malaria parasite as the pathogen and female *Anopheles* mosquitoes as the vector. It has a long epidemic history, seriously endangers human health, and affects social and economic development [1,2]. It is a major public health issue of common concern and is particularly a burden in developing countries. There were approximately 249 million cases of and 608,000 deaths caused by malaria worldwide in 2022, and 94% of global malaria cases and 95% of malaria deaths occurred in the African region. The number of global malaria deaths was almost the same as in 2021 and increased by 32,000 compared with 2019 [3].

The quinoline ring and endoperoxide bridge found in quinine and artemisinin, respectively, are part of the natural products contained in extracts of cinchona bark (quinine) and sweet wormwood (*Artemisia annua*), which have been used for centuries as herbal remedies to treat fevers [4]. Although quinine has remarkable curative effects, it also has side effects. With the deepening of research, scientists have developed chloroquine and amodiaquine, which play an important role in malaria treatment. Studies have found that the malaria parasite has developed serious resistance to traditional antimalarial drugs, such as quinine and chloroquine, which poses a challenge for malaria control [5]. The application of artemisinins has greatly contributed to malaria prevention and control. Artemisinin and its derivatives have high efficiency, quick effects, and low toxicity in malaria treatment and have become the first-choice treatment (especially for falciparum malaria). The World Health Organization (WHO) recommends artemisinin-based combination treatments (ACTs) as the first-line treatment for uncomplicated *P. falciparum* malaria and chloroquine-resistant P. vivax malaria [6].

The action mechanism of artemisinin and its derivatives has attracted much attention, as identifying the mechanism is critical to modifying the structure of artemisinins and improving their antimalarial efficacy. Although many studies have focused on the antimalarial mechanisms of artemisinin and its derivatives, the mechanism remains unclear [6,7]. The malaria parasite has developed resistance to artemisinin drugs, which hinders the prevention and treatment of malaria. Since its emergence in Cambodia, artemisinin resistance has also been found in many countries and regions in Southeast Asia [8,9,10,11,12], and artemisinin-resistant malaria parasites have also emerged in Africa [13,14]. In recent years, the research on artemisinin’s resistance mechanism has mainly focused on the Kelch13 (K13) mutation as K13 was the first gene found to be related to artemisinin resistance [15,16]. Some other genes were also found to be involved in resistance development. Understanding the antimalarial mechanism of artemisinin and its derivatives helps us develop artemisinin-based antimalarial medicines and guides the rational use and improvement of antimalarial drugs. Research on the prevalence and mechanisms of artemisinin resistance is conducive to a better understanding of how the malaria parasite is treatment-resistant to guide the development of new drugs and optimize existing treatment strategies, which is also helpful for malaria control and drug development. Therefore, this paper aims to review the actions of antimalarials and resistance mechanisms of artemisinin and its derivatives.

## 2. A Brief Introduction to Artemisinin and Its Derivatives

Traditional antimalarial drugs such as chloroquine, sulfadoxine, amodiaquine, and mefloquine have been widely used in malaria treatment. However, the resistance of *Plasmodium falciparum* to these drugs weakens their killing effect against the malaria parasite, which greatly challenges malaria prevention and control. A series of herbs, including artemisinin-containing plants, were used to treat fevers caused by malaria before the active ingredient was isolated [6]. A Chinese scientist, Tu Youyou, led her team to isolate and extract artemisinin from *Artemisia annua* for the first time in 1972. Artemisinin is a sesquiterpene lactone compound with a peroxy group and is composed of carbon, hydrogen, and oxygen. Its molecular formula is C_15_H_22_O_5_, which shows a good antimalarial effect.

The isolation of artemisinin brought new hope for malaria prevention and control. Artemisinin’s clinical value was initially limited by various deficiencies, such as its poor solubility, poor stability, low oral bioavailability, and short plasma half-life. A lactone ring in artemisinin’s molecular structure is formed by connecting two carbon atoms via a peroxy bond. This peroxy bond is key to the antimalarial action of artemisinin molecules [17]. Therefore, scientists modified its structure based on the peroxide bridge to maximize artemisinin’s efficacy. Various efficient semisynthetic artemisinin derivatives, such as dihydroartemisinin, artemether, and artesunate, have been developed (Figure 1). In addition, recent studies have shown the therapeutic potential of Platinum^IV^–artesunate complexes as antimalarial drugs [18]. With in-depth studies, artemisinin compounds have been widely used in clinical practice, and their clinical application is no longer limited to malaria treatment. They also have good application potential in the treatment of other parasitic diseases, tumors, inflammation, and other diseases [19,20,21,22].

## 3. Antimalarial Mechanism of Artemisinin and Its Derivatives

Artemisinin and its derivatives have potent and quick antimalarial action. They reduce the number of malaria parasites and relieve malaria symptoms by interrupting the life cycle of the erythrocytic stage of the malaria parasite, especially affecting the formation of merozoites in red blood cells. Understanding how they induce the malaria parasite’s death helps develop new artemisinin-based drugs. Key factors and targets of artemisinin and its derivatives of action are the hot spots of the current research (Figure 2).

### 3.1. Key Factors of Antimalarial Action of Artemisinin and Its Derivatives

Reactive oxygen species (ROS) are oxygen radical-containing or pro-radical molecules, including superoxide anion, hydroxyl radical, peroxy radical, hydrogen peroxide, and lipid hydroperoxide. ROS inhibit the survival of malaria parasites by attacking and destroying lipids, proteins, and nucleic acids [23]. Reduced iron in the unstable iron pool in the cytoplasm of *Plasmodium* and iron produced by heme decomposition can activate artemisinin to produce free radicals, which are extremely important in the antimalarial process. Two types of free radicals are mainly involved in the antimalarial effect, namely, oxygen free radicals and carbon free radicals. However, they are not produced simultaneously. Carbon radicals are formed based on oxygen free radical production. The peroxy bridge in the compound is broken under the influence of iron, followed by the production of oxygen free radicals and the formation of carbon free radicals via electron rearrangement [24].

### 3.2. Action Targets of Artemisinin and Its Derivatives

Proteins, lipids, and nucleic acids play pivotal roles in energy storage, cellular structure maintenance, and genetic information transmission for living organisms, including malaria parasites. These biomacromolecules collaborate to uphold the regular functioning and life cycle of an organism. Artemisinin medications disrupt the normal physiological functions of *Plasmodium* by targeting proteins, lipids, and nucleic acids, leading to the parasites’ death.

#### 3.2.1. Protein

Free radicals can impair the function of certain proteins and lead to the death of *Plasmodium* [25]. Translationally controlled tumor protein (TCTP) is one of the target proteins of free radicals. Artemisinin can interfere with the cytokine-like effect of TCTP, suggesting that artemisinin may kill *Plasmodium* parasites via TCTP [26]. Artemisinin and its derivatives can also specifically inhibit *Plasmodium falciparum* calcium ATPase 6 (*Pf*ATP6), which increases the calcium concentration in the *Plasmodium* cytoplasm and causes apoptosis [27]. It has been reported that dihydroartemisinin can down-regulate the expression of the Var gene encoding *Plasmodium falciparum* erythrocyte membrane protein-1 (*Pf*EMP1). Artemisinin sensitivity is significantly increased after *Pf*EMP1 knockout, suggesting that *Pf*EMP1 may be involved in the antimalarial mechanism of dihydroartemisinin [28]. Artemisinin can activate the endoplasmic reticulum stress response to damage proteins and hinder the degradation of damaged proteins by inhibiting proteasome, leading to the accumulation of polymerized proteins, thereby killing the parasite [29]. Previous studies have shown that artemisinin may disrupt *Plasmodium falciparum* DNA damage-inducible protein 1 (*pf*DDI1). The damaged pfDDI1 loses its ability to degrade the damaged protein, leading to the accumulation of the damaged protein and, ultimately, the parasite’s death [30]. In addition, Priya Arora et al. [31] found that *Plasmodium falciparum* ubiquitin-specific protease (*Pf*USP) knocked-out parasites showed increased sensitivity to dihydroartemisinin, suggesting that *Pf*USP plays an important role in dihydroartemisinin’s antimalarial effect. The results of Gao et al.’s study suggest artemisinin can bind to parasite proteins via both covalent and non-covalent modification, and these might jointly contribute to its antimalarial effect [32].

#### 3.2.2. Lipids

Polyunsaturated fatty acids (PUFAs) are important phospholipids in the membrane structure that control the fluidity and structural transformation of the cell membrane [33]. High PUFA levels are vulnerable to ROS. Artemisinin and its derivatives damage the membrane structure and reduce the membrane fluidity of the malaria parasite, such as by stratifying the cytomembrane, food vacuole membrane, and nuclear membrane; the swelling and separation of the outer and inner membrane; or shrinking the mitochondria [34]. Artemisinin increases ROS levels, reduces the membrane potential of *Plasmodium* mitochondria, and affects its membrane permeability. This effect is irreversible, thus damaging the mitochondria to achieve an antimalarial effect [35]. Dihydroartemisinin treatment significantly changes the content of some lipids in the plasma membrane of *Plasmodium falciparum* [36], while most of these lipids contain polyunsaturated fatty acids vulnerable to ROS [37]. Therefore, the mechanism underlying artemisinin’s effect on the membrane structure may be related to lipid peroxidation. Furthermore, the lipid peroxidation product 4-hydroxynonenenal (4-HNE) was produced in human erythrocytes infected with *Plasmodium falciparum* after dihydroartemisinin treatment [38]. Iron death is a form of cell death, which is characterized by excessive ion accumulation in cells, leading to lipid peroxidation, cell membrane rupture, and cell death. ROS play a key role in iron death [39]. The antimalarial effect of dihydroartemisinin is additive or synergistic when it is combined with an iron death inducer, while its antimalarial effect is inhibited when combined with an iron death inhibitor [40]. Therefore, artemisinin and its derivatives may damage the membrane structure of *Plasmodium* via lipid peroxidation to achieve an antimalarial effect.

#### 3.2.3. Nucleic Acid

The molecular integrity of DNA and other genetic materials is necessary for the persistence and survival of organisms. DNA modification is mainly accomplished via the 8-hydroxylation of guanine. Due to their low oxidation potential, ROS cause structural damage to DNA mainly by attacking guanine [41]. The double strands of the DNA of the *Plasmodium* parasite were broken after artesunate treatment, which was related to the increase in ROS production [42]. According to Gunjan S. [43], ROS produced by artemisinin derivatives’ activation can depolarize the mitochondrial membrane potential of *Plasmodium* and activate the aspartic acid proteolytic enzyme containing cysteine to cause DNA fragmentation, suggesting this leads to the apoptosis and death of *Plasmodium*.

## 4. Artemisinin Resistance

Clinically, artemisinin resistance refers to the delayed clearance of malaria parasites in the peripheral blood of patients treated with an artemisinin derivative or an artemisinin combination therapy (ACT) [44]. It is more appropriate to refer to this delayed clearance as “partial resistance” to emphasize this time-limited and cycle-specific feature, as studies have shown that the resistance mechanisms developed by the parasites against artemisinin compounds only affect the ring stage of the malaria parasite cycle in humans [45]. It is worrying that partial artemisinin resistance might further evolve to affect other stages of the parasites, developing into complete resistance.

Since artemisinin resistance was discovered in western Cambodia in 2008, artemisinin resistance has been found in Laos, Myanmar, Thailand, Vietnam, and other countries, especially in the Mekong subregion [8,9,10,11,12]. In addition to Southeast Asia, artemisinin-resistant malaria patients have been reported in African regions, such as Rwanda [13,14], the Democratic Republic of the Congo, Ethiopia [46], and Eritrea [47]. Partial artemisinin resistance remains a threat in the African region (especially in Eritrea, Rwanda, and Uganda), the Greater Mekong subregion (GMS), South America, and Papua New Guinea [3]. In some African countries such as Uganda, ACTs are still efficacious in malaria treatment, although partial artemisinin resistance has been confirmed [48]. To prevent artemisinin resistance, an efficacious partner drug of ACTs is recommended, and the possibility of treatment failure caused by ACT partner drug should be considered [49,50]. Resistance to the ACT partner drugs has been found in the GMS [51]. Managing the drug resistance situation requires close efficacy monitoring for artemisinins and their partner drugs and a comprehensive understanding of the resistance mechanism. Several mechanisms are proposed to play a role in the development of artemisinin resistance in the *Plasmodium* parasite. Mutations in K13 and other genes are considered to lead to resistance development (Figure 3).

### 4.1. Mechanism of Artemisinin Resistance Mediated by Kelch13

Artemisinin resistance is mainly related to a mutation in the propeller domain of the gene encoding the K13 protein. K13 mutation leads to the decreased sensitivity of *Plasmodium* to artemisinin by blocking the activation of artemisinin and enhancing the antioxidant function of malaria parasites, reducing the damage of proteins, and enhancing the repair of DNA damage. It has been found that K13 mutation affects cell cycle periodicity, unfolded protein response, protein degradation, vesicle transport, and mitochondrial metabolism [52]. During the growth and development of red blood cells, *Plasmodium* consumes hemoglobin and utilizes amino acids. The breakdown of hemoglobin produces heme, which plays an important role in the activation of artemisinin. The K13 protein is related to the endocytosis of hemoglobin by the malaria parasite. *Pf*K13 mutation reduces the endocytosis of hemoglobin and the activation of artemisinin, resulting in decreased sensitivity to artemisinin [53]. In addition, K13 mutations may lead to the delayed clearance of malaria parasites via an increased antioxidant capacity. The up-regulation of the genes of the reactive oxidative stress complex involved in the unfolded protein response in *Plasmodium* could tolerate protein damage [54]. Furthermore, K13 mutation is related to the up-regulation of the stress oxidative stress complex gene [52]. The malaria parasite enhances DNA repair when K13 mutation occurs [55].

### 4.2. Prevalence of Kelch13-Related Drug-Resistant Strains

In 2020, the WHO published 10 validated genetic loci related to artemisinin resistance, F446I, N458Y, M476I, Y493H, R539T, I543T, P553L, R561H, P574L, and C580Y [56], which were detected in many countries and regions (Table 1). The mutations were predominantly detected in the GMS and Africa. C580Y was one of the most common mutations in all Mekong subregion countries and China [57]. The C580Y K13 mutant allele has emerged independently in Guyana, a country in South America, with an evolutionary origin distinct from Southeast Asia based on genome analysis [58,59]. The mutation has not been identified in more recent samples in Guyana and may have disappeared.

### 4.3. Other Artemisinin Resistance-Related Genes

Not all artemisinin resistance was caused by K13 mutation, as artemisinin resistance has developed in some malaria parasites without K13 mutation [68,69]. *Plasmodium falciparum* actin-binding protein coronin (*Pf*coronin) plays a role in endocytosis. Therefore, *Pf*coronin mutation may cause the activation of artemisinin to be blocked and its activation products to be reduced, resulting in a reduced killing effect of artemisinin against the malaria parasite. *Pf*coronin mutants may appear as non-kelch13 resistance in the natural environment [70]. For example, *Pf*coronin is associated with a reduced sensitivity of *Plasmodium* to artemisinin in Nigeria [71]. In addition, adaptor protein 2μ (AP-2μ) and upstream binding protein 1 (UBP1) are involved in the endocytosis of hemoglobin with K13 [53]. AP-2μ and UBP1 mutations may affect the uptake of hemoglobin by *Plasmodium* parasites, which may affect the activation of artemisinin, resulting in a decrease in the production of ROS, leading to artemisinin resistance. AP-2μ and UBP1 mutations reduce the in vitro artemisinin sensitivity of the early ring of *Plasmodium falciparum* [72]. AP-2μ and UBP1 mutations were detected in malaria patients imported from Africa and Southeast Asia to Wuhan, China [73]. *Pf*UBP1 and *Pf*AP-2μ gene mutations were detected in all isolates of *Plasmodium falciparum* in Bioko Island, Equatorial Guinea, in which the prevalence of *Pf*UBP1 gene mutation was lower and that of *Pf*AP-2μ gene mutation was higher [74]. After ACT treatment, single-nucleotide polymorphisms in *Pf*AP-2μ and *Pf*UBP1 genes were associated with the delayed clearance of malaria parasites in Kenya and recurrent imported malaria in Ghanaian isolates [75]. Furthermore, *Pf*UBP1 mutations were also found in Myanmar [76]. In addition to the validated genes, non-synonymous polymorphisms of ferredoxin, apicoplast ribosomal protein S10, multidrug resistance protein 2, and chloroquine resistance transporter are also closely associated with artemisinin resistance [77], but this association needs further verification.

## 5. Discussion

Since the isolation of artemisinin, the antimalarial mechanism of artemisinin has been a research hotspot, and many possible mechanisms have been proposed. However, many scientific issues need to be further studied. It is now widely accepted that artemisinin exerts antimalarial effects by generating free radicals, which kill malaria parasites by acting on proteins, nucleic acids, and lipids. Moreover, multiple proteins of malaria parasites have been recognized as targets of artemisinin. However, the underlying mechanisms remain unclear. Many studies show that artemisinin-based drugs can change the contents of some lipids in the *Plasmodium* membrane [35,36,38], but related research on non-plasma membrane lipids is still lacking. This research could help develop new drugs to enhance artemisinin’s antimalarial effect and further study the action mechanism of artemisinin drugs. Drug resistance to artemisinin, a key drug for malaria treatment, poses a great challenge to malaria control. Artemisinin resistance seriously affects the efficacy of artemisinin-based drugs.

Artemisinin resistance has been widely investigated in recent years, but the resistance mechanism needs further study. Most studies on artemisinin resistance have focused on K13 mutations; however, the mechanism of action of K13 mutations on artemisinin resistance needs more in-depth and comprehensive research, for example, regarding whether iron content changes after K13 mutation (which plays a key role in the activation of artemisinin) and whether it can also change artemisinin’s damage effect on lipids. In addition to K13, *Pf*AP2μ, UBP1, and *Pf*coronin, others have also been associated with artemisinin resistance. Current studies indicate their involvement in *Plasmodium* endocytosis. However, further research is needed to determine whether they directly impact artemisinin activation. Additionally, it remains unclear if these genes employ alternative mechanisms to inhibit the action of artemisinin and how they affect its damage to the proteins, nucleic acids, and lipids in the malaria parasite. Therefore, a more comprehensive and in-depth investigation is required to elucidate the roles of K13, *Pf*AP2μ, UBP1, and *Pf*coronin in artemisinin resistance. Furthermore, new resistance-related genes, in addition to the verified K13, AP-2μ, UBP1, and coronin, require validation and exploration to guide malaria control and drug development.

Novel antimalarials need to be developed urgently due to the serious situation of malaria prevalence and the challenge of artemisinin resistance. A comprehensive understanding of the antimalarial and resistance mechanisms associated with artemisinin can inform and enhance drug development strategies. Scientists have been developing novel antimalarials in recent years. Fully synthetic endoperoxides have been extensively studied, some of which are currently in clinical and preclinical trials. However, some compounds have shown various shortcomings. For example, the clinical development of antimalarials, such as OZ439 OZ277 (127b), OZ439 (135a), OZ657 (135b), OZ493 (136a), OZ609 (136b), and OZ655, was recently halted due to formulation challenges stemming from the drug’s lipophilicity and low aqueous solubility [78]. Fortunately, recent research on artemisinin optimization suggests that Platinum^IV^–artesunate multi-action prodrugs may be a promising antimalarial drug candidate [18].

In summary, there is considerable uncertainty about the antimalarial mechanism of artemisinin and its derivatives, which is extremely disadvantageous to the research and development of antimalarial drugs, drug application, and malaria control. Therefore, further studies are needed to explain the mechanism of artemisinin drugs in malaria treatment. In addition, elucidating the antimalarial mechanism of artemisinin drugs will help us to study the molecular mechanism of resistance to malaria parasites. It can also promote an improvement in artemisinin drugs and the research and development of new antimalarial drugs to explore their role in blocking malaria transmission.

## Figures and Tables

**Figure 1 tropicalmed-09-00223-f001:**
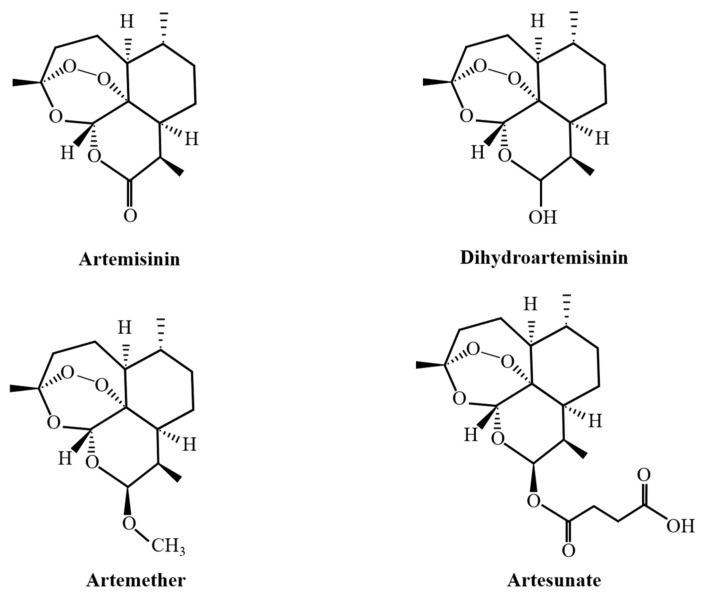
Chemical structures of artemisinin and its derivatives.

**Figure 2 tropicalmed-09-00223-f002:**
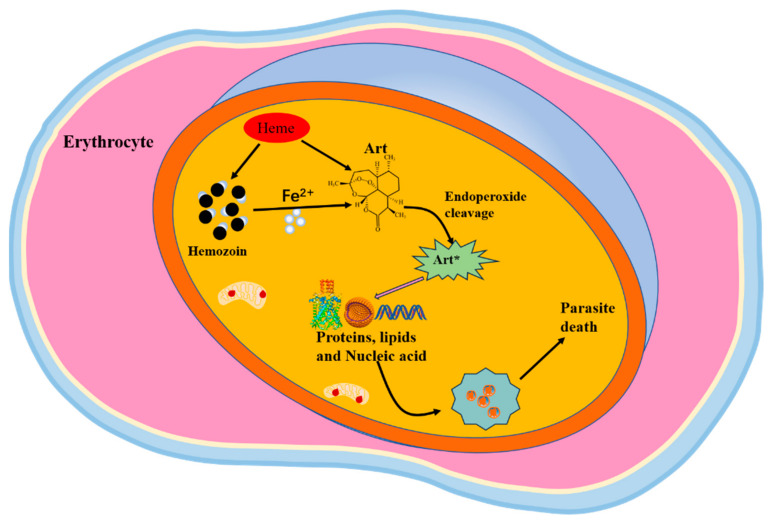
The antimalarial mechanism of artemisinin. Heme and iron produced by hemozoin can activate artemisinin to produce free radicals. The activated artemisinin disrupts the physiological functions of *Plasmodium* by targeting proteins, lipids, and nucleic acids, leading to the parasites’ death. The asterisk signifies the activated form of ART. Abbreviations: Art, artemisinin.

**Figure 3 tropicalmed-09-00223-f003:**
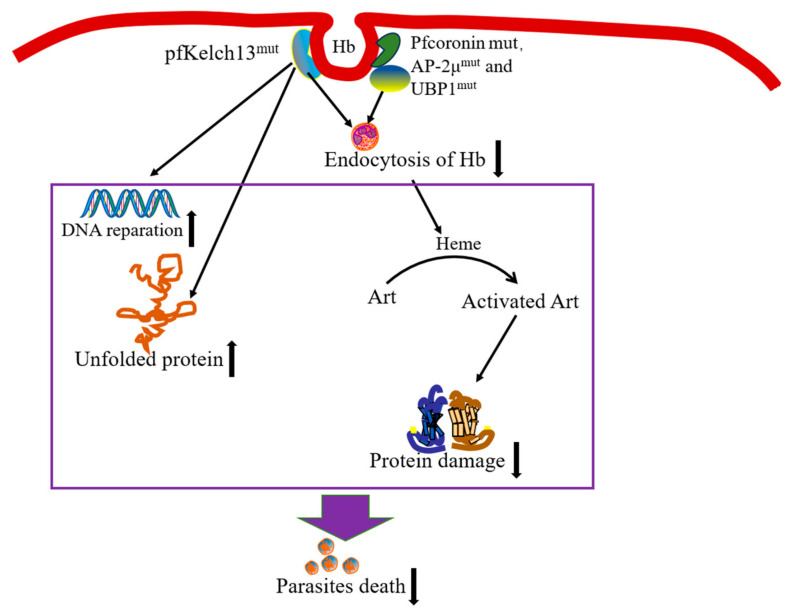
The mechanism of artemisinin resistance. The *Pf*K13 mutation causes the reduced endocytosis of host hemoglobin, resulting in lowered levels of hemoglobin catabolism and the reduced activation of artemisinin drugs. The K13 mutation also leads to increased resistance to artemisinin by reducing the damage of proteins and enhancing the repair of DNA damage. In addition, *Pf*coronin, AP-2μ, and UBP1 mutations may block artemisinin activation, resulting in a reduced killing effect of artemisinin. Abbreviations: ART, artemisinin; Hb, hemoglobin; *Pf*coronin, *Plasmodium falciparum* actin-binding protein coronin; AP-2μ: adaptor protein 2μ; UBP1, upstream binding protein 1.

**Table 1 tropicalmed-09-00223-t001:** The distributions of 10 validated genetic loci related to artemisinin resistance in isolates from different countries and geographic regions.

Drug-Resistant Strain	Country
C580Y	Thailand [9,60], Laos [10], Vietnam [11], Myanmar [12], Papua New Guinea [61], and China [62]
P574L	Vietnam [11] and Thailand [9]
F446I	India [63], Myanmar [64], and China [65]
N458Y	Thailand [9], Myanmar [12], and China–Myanmar border [16]
M476I	Myanmar [12], Thailand [66], and Africa [46]
Y493H	Cambodia, Laos, eastern Thailand, and Vietnam [10,11]
I543T	Cambodia, Laos, Thailand, and Vietnam [57]
P553L	Vietnam [67], Ghana, Angola, Equatorial Guinea, and Nigeria [46]
R561H	Thailand [60], Myanmar [64], and Rwanda [13,14]

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
