# Peer review of "Antimalarial Mechanisms and Resistance Status of Artemisinin and Its Derivatives"

_tropicalmed, 2024, doi:10.3390/tropicalmed9090223_

Round 1

Reviewer 1 Report

Comments and Suggestions for Authors

This article provides a review of the proposed mechanisms of action of artemisinins and artemisinin resistance. While some interesting information is presented, the article requires significant revision before it can be considered for publication.  I have listed a few points below to assist with the revision.

General:

1.     I recommend an English editor reviews the article before resubmission.

2.     Please ensure terms are written out in full before abbreviations are used and once the abbreviation is introduced, please use it.

3.     Please ensure you have references for statements of fact made throughout this manuscript.

Abstract:

1.     Ln12: Why is a discussion on the shortcomings of artemisinin presented here?

2.     Ln14-15: I do not understand the point being made here.  Please rephrase for clarity.

3.     Ln17: replace “has” with “have”

4.     Ln18: Do not understand what “clinical application” means.  Please provide clarity.

5.     Ln20: Artemisinin resistance has not been confirmed across the globe.  Please correct this statement.

6.     Ln20-21: Please clarify how understanding resistance mechanisms help with malaria control/elimination efforts.

7.     Ln22: Please clarify how understanding resistance mechanisms helps guide the deployment of antimalarials.

8.     Ln24: Do not understand how understanding resistance mechanisms will prevent the emergence of antimalarial drug resistance. Please provide clarity

Introductions:

1.     Ln31: Are there other vectors of malaria?

2.     Ln31-32: A reference is needed for this statement

3.     Ln33: Malaria is not a problem across the globe – it is a burden in the developing world.  Please clarify this.

4.     Ln35: Suggest you provide more detail on where the burden is highest.

5.     Ln40: Reference for this statement is needed.  Expand on why there is so much research on identifying resistance mechanisms.

6.     Ln21: Delete “at the same time, in recent years”

7.     Ln46: Why is artemisinin-resistance in Africa more serious than in SE Asia?

8.     Ln47-48: Do not understand the point being made.  Please clarify.

9.     Ln48-49: You need to explain why the kelch13 gene is so well researched.

10.  Ln53: Provide more information on how understanding resistance mechanisms helps to control malaria.

11.  Ln55: Assume you are going to discuss actions of antimalarials and mechanisms of resistance – right? Please clarify.

Artemisinins and derivatives:

1.     Ln57: sulfamethoxine has not been used in malaria control.  I am assuming you mean “sulfadoxine”.  Please correct the spelling of amodiaquine.

2.     Ln59: I assume you mean killing effect on the parasite and not the host. Please clarify.

3.     Ln61: Artemisinins containing plants were used to treat fevers/malaria well before the active ingredient was isolated.  Please state this clearly.

4.     Ln65: not birth – rather isolation or identification.

5.     Ln66: Please clarify what you mean by “many deficiencies in the process of clinical application.”

6.     Ln68: Expand on why the short half-life is a limitation.

7.     Ln77: What does “deepening research” mean?

8.     Ln78-79: Do you mean that the artemisinins are being investigated as potential treatments for other diseases?

Mechanism of action of artemisinins:

1.     Ln83: More interrupting than interfering.

2.     Ln86: Do you mean new artemisinin-based drugs?

3.     Ln88-90: Please rephrase this sentence for clarity.

4.     Ln90: Do you mean ROS increases the survival rate of malaria parasites?

5.     Ln91: How does this activation occur? What causes the activation?

6.     Ln96: Not sure what point is being made here. Please clarify.

7.     Ln99: Please provide some context here before going into the subsections.

8.     Ln101: Do free radicals impair function in all proteins or only certain proteins?

9.     Ln102-103: Do you mean free radicals target the TCTP?

10.  Ln117: What has chloroquine got to do with the discussion here?

11.  Ln118-119: Do not understand what point is being made here.

12.  Ln122: Is this similar to pfDDI1 action?

13.  Ln125: What do you mean by “deformation”

14.  Ln127: Swelling of what? Widening the gap of plasms membrane and what?

15.  Ln133-137: Unclear what points are being made here. Please rephrase this section for clarity.

Artemisinin resistance:

1.     Ln162: Great pains are taken to list all countries in SE Asia effected by artemisinin resistance, the same respect should be shown to the African countries.

2.     Ln165: Are you suggesting the kelch13 mutations enhance resistance that is already present?

3.     Ln170: What do you mean by “devours hemoglobin”

4.     Ln172: Are you suggesting hemoglobin breakdown is not required for parasite survival?

5.     Ln182: Rather validated than certified.

6.     Ln198-199: Provide a reference to support this statement.

Discussion:

1.     Ln224: Discovery/isolation better than advent

2.     Ln226-227: What do you mean here?

3.     Ln229: What does “lack specificity” mean?

4.     Ln236: Artemisinin resistance affects the efficacy of artemisinin-based drugs and not all antimalarials.  Please clarify.

Comments on the Quality of English Language

Comments included in my comments to the authors above.

Reviewer 2 Report

Comments and Suggestions for Authors

Dear authors,

I found your paper interesting and could be useful for the new researchers and beginners . 

Please see attached file for comments for revision.

Best wishes .

Comments on the Quality of English Language

I didn't find  any major issues in the English language .

Reviewer 3 Report

Comments and Suggestions for Authors

Dear Author(s)

The authors tried to gather some articles containing antimalarial mechanisms and resistance status of artimisinin and its derivatives in the format of a systematic review study to provide a reference for new relevant studies.

Although your manuscript is useful text, the category  of the manuscript is  really unknown. Neither original study nor systematic review. Indeed, the manuscript looks like a normal scientific report in the field of the entitled topic .

Although the manuscript consists of some relevant articles, did not result in a significant out come.

The study did not follow a reliable methodology.

The manuscript was concluded that;"there is still a lot of uncertainty about the antimalarial mechanisms of artimisinin and its derivatives... therefore, further studies such as molecular studies are needed to explain the relevant mechanisms".This conclusion is in contrary to the aim that has been mentioned in the introduction of the manuscript.

The references are appropriate.

Comments on the Quality of English Language

Quality of English Language is good.

Reviewer 4 Report

Comments and Suggestions for Authors

The article "Antimalarial Mechanisms and Resistance Status of Artemisinin and Its Derivatives" has been reviewed. The study discusses the antimalarial mechanisms and resistance status of artemisinin and its derivatives, offering valuable insights for mitigating drug resistance and the development of novel antimalarial drugs. However, it requires significant revision before further consideration.

1. The introduction section should provide a concise overview of malaria and its challenges in chemotherapy, supported by relevant statistical data. Given that the review emphasizes the medicinal chemistry of antimalarial drugs, this information is crucial.

2. It is essential for the article to be written in a manner that captivates not only experts in the field but also a general audience.

3. Including chemical numbering in artemisinin in Figure 1 would enhance clarity and facilitate better understanding.

4. The authors should focus on creating figures that are not only informative but also captivating, encouraging readers to peruse the entire article or at least engage with the figures to gain new insights. Incorporating a schematic figure could enhance reader engagement.

5. The authors should consider how readers can derive substantial value from the tables, such as listing all plant species of Artemisia and their uses, without having to delve into the text extensively. Sustaining reader interest beyond two-thirds of the figures and tables is crucial.

6. A comprehensive elucidation of the mechanism of action of artemisinin is warranted.

7. The number of references provided is insufficient, and the authors should incorporate recent studies to expound on their points.

8. It is advisable to lower the similarity index to below 20%. The current manuscript exceeds 40% in similarity index.

Comments on the Quality of English Language

Moderate editing of English language required.

Reviewer 5 Report

Comments and Suggestions for Authors

This comprehensive and well-written review of the current understanding of artemisinin resistance requires minimal revision. For suggested edits, please see notes included in the attached pdf. 

Comments on the Quality of English Language

In general, the quality of English language is very high. I have made some suggestions for improvement in the attached pdf. However, the paper would be helped by having a native English speaker to a last review, especially the discussion section where the writing seems rushed and less polished than the rest of the paper. 

Reviewer 6 Report

Comments and Suggestions for Authors

Review Report for tropicalmed-3118839

Dan et al. have written a short review summarizing the antimalarial mechanisms of artemisinin and its derivatives, as well as their associated drug resistance issues. This could serve as a useful beginner's guide for new researchers in the field. While the manuscript aligns with the journal's readership, the current version is very simple and not acceptable in its present form. Below are some suggestions for improvement:

1. The structure of artemisinin and its derivatives should be included.

2. The manuscript lacks figures, which makes it less accessible to readers. I recommend that the authors present the mechanisms in a more graphic manner.

3. The manuscript should include the most recent work on artemisinin optimization, such as J. Med. Chem. 2023, 66, 12, 8066–8085.

4. Several recent studies on target identification for artemisinin and its derivatives are both interesting and important; these should be included and well described in the manuscript. For example, see Chinese Chemical Letters, 2023, 34, 12, 108296.

5. The latest clinical developments on drug resistance should be discussed in greater detail.

6. In the introduction, past antimalarial drugs, such as quinine, should be mentioned.

7. In the discussion section, new promising drug candidates should be highlighted.

Comments on the Quality of English Language

title 3.2, line 99, align the font.

Round 2

Reviewer 3 Report

Comments and Suggestions for Authors

Dear Author(s)

Thank you for your acceptable response to my comments.

Author Response

Thanks for your comment!

Reviewer 4 Report

Comments and Suggestions for Authors

The authors made significant revisions based on the reviewers comments the MS can be accepted for publication.

Comments on the Quality of English Language

Minor editing of English language required.

Author Response

Thanks for your comment and suggestion!

We have checked and revised the English expression throughout the manuscript seriously. The revisions were highlighted with yellow colour.  

Reviewer 6 Report

Comments and Suggestions for Authors

Review Report for tropicalmed-3118839-2

This manuscript has been well improved. All my concerns have been addressed. However, two more minor issues should be further addressed before publication: (1) The structure in Figure 1 should be further revised by a professional chemist, some bonds are not pretty, and even not properly attached to the carbon; the structure of artesunate and dihydroartemisinin are incorrect, the bold bond at C-6 should be a hydrogen rather than methyl. (2) Reference No 23, page information needs to be included. I Suggest to accept this manuscript for publication once these two issues are resolved.

Author Response

Thanks for comment and suggestions!

The figure 1 has been revised seriously under the help of a professional chemist.

ALL the references have been checked and the lacking page information has been added.